# Lef1 expression in fibroblasts maintains developmental potential in adult skin to regenerate wounds

Quan M Phan[1], Gracelyn M Fine[1], Lucia Salz[1], Gerardo G Herrera[1], Ben Wildman[1], Iwona M Driskell[1], Ryan R Driskell[1,2]*

[1]School of Molecular Biosciences, Washington State University, Pullman, United States; [2]Center for Reproductive Biology, Washington State University, Pullman, United States

**Abstract** Scars are a serious health concern for burn victims and individuals with skin conditions associated with wound healing. Here, we identify regenerative factors in neonatal murine skin that transforms adult skin to regenerate instead of only repairing wounds with a scar, without perturbing development and homeostasis. Using scRNA-seq to probe unsorted cells from regenerating, scarring, homeostatic, and developing skin, we identified neonatal papillary fibroblasts that form a transient regenerative cell type that promotes healthy skin regeneration in young skin. These fibroblasts are defined by the expression of a canonical Wnt transcription factor Lef1 and using gain- and loss of function genetic mouse models, we demonstrate that Lef1 expression in fibroblasts primes the adult skin macroenvironment to enhance skin repair, including regeneration of hair follicles with arrector pili muscles in healed wounds. Finally, we share our genomic data in an interactive, searchable companion website (https://skinregeneration.org/). Together, these data and resources provide a platform to leverage the regenerative abilities of neonatal skin to develop clinically tractable solutions that promote the regeneration of adult tissue.

*For correspondence:
ryan.driskell@wsu.edu

Competing interests: The authors declare that no competing interests exist.

## Introduction

Understanding how to induce skin regeneration instead of scarring will have broad implications clinically and cosmetically (*Walmsley et al., 2015b*). One of the main characteristics of scars is the absence of hair follicles, indicating that their regeneration in a wound may be a critical step in achieving scar-less skin repair (*Yang and Cotsarelis, 2010*). Interestingly, human embryonic skin has the capacity to regenerate without scars (*Lo et al., 2012*). Similarly, neonatal and adult mouse skin has the capacity to regenerate small non-functional hair follicles under specific conditions (*Figure 1c–d*; *Ito et al., 2007*; *Rognoni et al., 2016*; *Telerman et al., 2017*). These insights have prompted efforts in the field to define the molecular triggers that promote hair development in skin, with the ultimate goal of devising a way to regenerate fully functional hairs in adult skin wounds as a therapeutic modality (*Yang and Cotsarelis, 2010*).

Human and mouse skin are similar in their overall structural complexity, indicating that mouse skin can be a useful model to study skin development and wound repair (*Chen et al., 2013*). Murine hair follicle and skin development primarily occurs between embryonic day 12.5 (E12.5) to post-natal day 21 (P21) (*Müller-Röver et al., 2001*). During this time, different fibroblast lineages are established that respond to the changes in the environment to support hair follicle and skin development (*Driskell et al., 2013*; *Jiang et al., 2018*; *Rinkevich et al., 2015*; *Rognoni et al., 2016*). Fibroblasts that support hair follicle development differentiate from the papillary fibroblast lineage, into dermal papilla (DP), dermal sheath (DS), and arrector pili (AP) cells (*Driskell et al., 2013*; *Plikus et al., 2017*). Reticular fibroblasts, which cannot differentiate into papillary fibroblast lineages, secrete

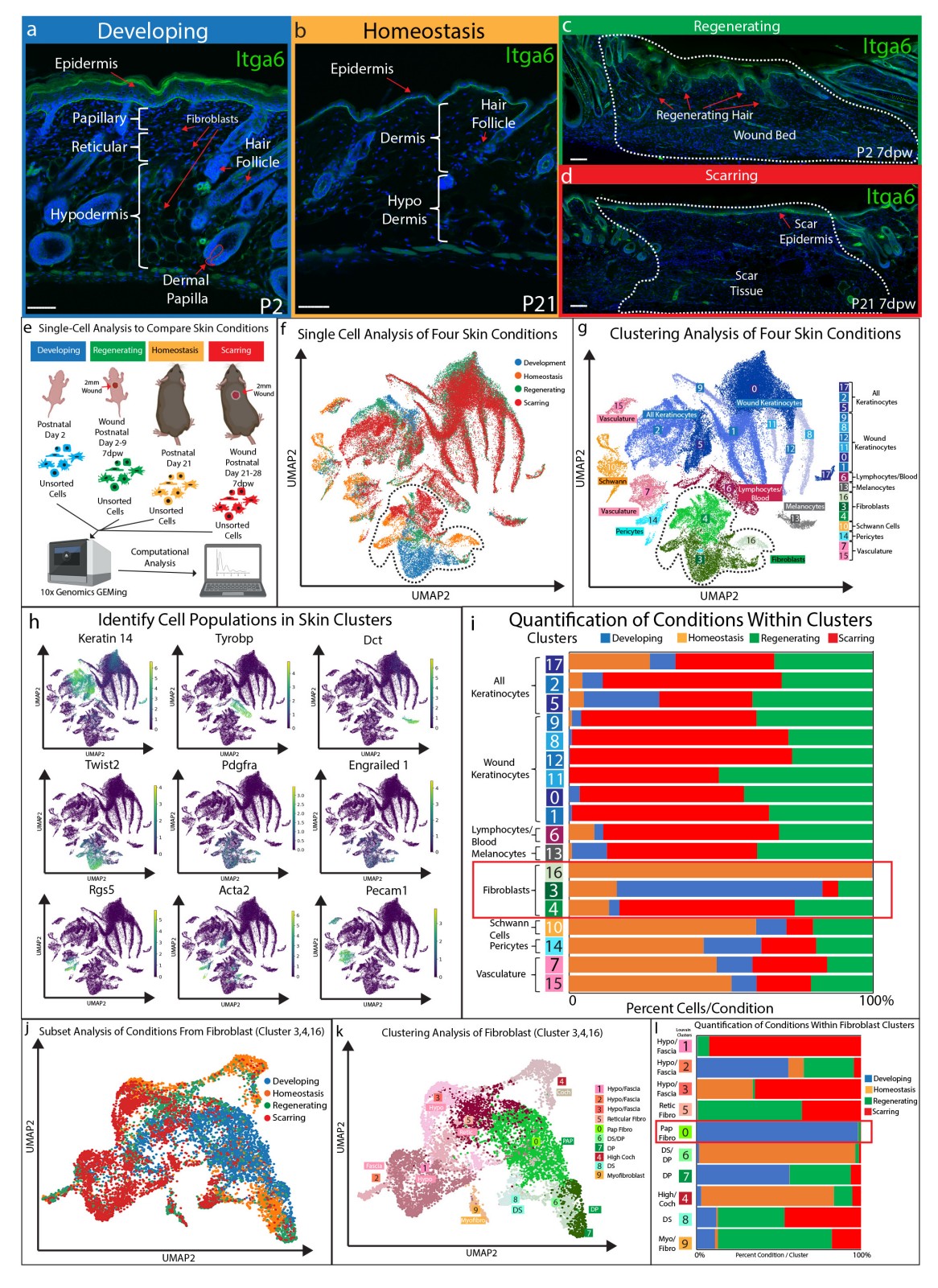

**Figure 1.** Fibroblasts in developing skin are distinct from regenerating, scarring, and homeostatic conditions. (**a–d**) Immunostaining analysis and histological map of the dermis in developing Post-natal day 2 (P2) skin (**a**), homeostatic skin at P21 (**b**), skin undergoing regeneration P2 7 days post wound (7dwp) (**c**), and skin repairing a wound via scarring P21 7dpw (**d**). (**e**) Schematic representation of single-cell isolation, library preparation, and sequencing analysis for unsorted cells from developing (P2), regenerating (P2 7dpw), homeostatic (P21), and scarring (P21 7dpw) conditions. (**f**) UMAP

*Figure 1 continued on next page*

*Figure 1 continued*

visualization of all cell populations for all conditions. Each cell is color coded based according to condition as labeled. (g) Clustering analysis of the UMAP plot, color coded based on cell types. (h) Overlaying gene expression on UMAP clusters to identify cell types. (i) Quantification of the percentage of all cells represented within a cluster. (j) Subset and re-clustering of Clusters 3, 4, and 16 by computational integration. (k) Cluster analysis of integrated fibroblast clusters 3, 4, and 16. (l) Quantitation of fibroblasts within each cluster represented by condition. n = 3 biological replicates for each condition. Each biological replicate was used to generate a single library. Scale bars are 100 μm.

The online version of this article includes the following figure supplement(s) for figure 1:

**Figure supplement 1.** Classifying fibroblast clusters and conditions in development, regeneration, scarring, and homeostasis.

**Figure supplement 2.** Batch analysis of biological replicates for developing, homeostatic, regenerating, and scarring conditions.

extra-cellular-matrix (ECM) and form adipocytes (*Driskell et al., 2013*; *Schmidt and Horsley, 2013*). By post-natal day 2 (P2) fibroblast heterogeneity is fully established with the presence of the defined layers of the dermis (*Figure 1a*; *Driskell et al., 2013*). Skin maturation occurs after P2 with the formation of the AP and the completion of the first hair follicle cycle, which results in the loss of a defined papillary fibroblast layer (*Figure 1b*; *Driskell et al., 2013*; *Rognoni et al., 2016*; *Salzer et al., 2018*). We have previously shown that papillary fibroblasts are the primary source of de-novo dermal papilla during skin development, which are required for hair formation (*Driskell et al., 2013*). Furthermore, it has been suggested that adult murine skin form scars due to the lack of a defined papillary layer (*Driskell et al., 2013*; *Driskell and Watt, 2015*). Consequently, expanding this fibroblast layer in adult skin might support skin regeneration in adult mice.

The use of scRNA-seq in the murine skin has established useful cell atlases of the skin during development and homeostasis (*Guerrero-Juarez et al., 2019*; *Haensel et al., 2020*; *Joost et al., 2020*; *Joost et al., 2018*; *Joost et al., 2016*; *Mok et al., 2019*). In addition, scRNA-seq studies investigating wound healing have so far focused on comparing scarring, non-scarring, or regenerating conditions (*Guerrero-Juarez et al., 2019*; *Haensel et al., 2020*; *Joost et al., 2018*). These studies have helped to identified key markers of the newly discovered skin fascia, including *Gpx3*, *Plac8*, and *Mest*, which recently has been shown to contribute to scar formation (*Correa-Gallegos et al., 2019*; *Grachtchouk et al., 2000*; *Joost et al., 2020*). These, scRNA-seq studies have revealed that transgenic activation of the Shh pathway in the alpha-smooth-actin cells in scarring wounds, which includes pericytes, blood vessels, and myofibroblasts, can support small non-functional hair regeneration (*Lim et al., 2018*). However, activation of Shh pathway in dermal fibroblasts is associated with malignant phenotypes and may perturb development and homeostasis such that it may not be a safe target to support human skin regeneration clinically (*Fan et al., 1997*; *Grachtchouk et al., 2000*; *Oro et al., 1997*; *Sun et al., 2020*). Altogether, these findings suggest that an overall comparison of developing, homeostatic, scarring, and regenerating skin conditions will yield important discoveries for molecular factors that can safely support skin regeneration without harmful side effects.

The Wnt signaling pathway is involved in regulating development, wound healing, disease and cancer (*Nusse and Clevers, 2017*). Studies that activated Wnt and beta-catenin in skin have led to important discoveries for wound healing but have produced contrasting results in the context of fibroblast biology and hair follicle formation (*Chen et al., 2012*; *Enshell-Seijffers et al., 2010*; *Hamburg-Shields et al., 2015*; *Lim et al., 2018*; *Rognoni et al., 2016*). Wnts are a group of secreted protein that activates a cascade of events that stabilizes nuclear beta-catenin, which operates as a powerful co-transcription factor of gene expression. Importantly, beta-catenin alone cannot activate the expression of Wnt target genes without co-transcription factors. There are four Wnt co-transcription factors Tcf7 (Tcf1), Lef1, Tcf3 (Tcf7l1), and Tcf4 (Tcf7l2). These co-transcription factors modulate the functional outcome of Wnt signaling by binding to different target genes (*Adam et al., 2018*; *Nguyen et al., 2009*; *Yu et al., 2012*). In the context of wound healing and regeneration, it is not known how differential expression of Tcf/Lef can modulate fibroblast activity via Wnt signaling.

Since it has been shown that embryonic and neonatal skin have the potential to regenerate hair follicles upon wounding (*Hu et al., 2018*; *Rognoni et al., 2016*), we set out to identify the cell types and molecular factors that define this ability in order to transfer this regenerative potential to adult

tissue. Our work has identified *Lef1*, as the factor in fibroblast of developing skin, that can transform adult tissue to regenerate.

## Results

### Developing fibroblasts are distinct cells that are associated with the ability to support hair follicle regeneration

We and others have previously shown that there are differences between neonatal and adult skin in terms of their cellular and biological properties (*Ge et al., 2020*; *Rognoni et al., 2016*; *Salzer et al., 2018*). These differences influence how skin can repair or if it can regenerate hair follicles in a wound. We showed that neonatal developing skin at P2 (*Figure 1a*) can regenerate small hair follicles 7 days after wounding (7dpw) (*Figure 1c*), while adult homeostatic skin at P21 (*Figure 1b*) heals by scarring 7dpw (*Figure 1d*). In order to identify the cell types and molecular pathways that define the ability of developing skin to regenerate, we performed a comparison of unsorted cells from four conditions using scRNA-seq (*Figure 1e*). A total of 66,407 cells from 12 libraries (n = 3 for each condition) met the preprocessing threshold, showed little batch variation (*Figure 1—figure supplement 2*), and were used for downstream analysis. Each condition was represented as different colors (*Figure 1e–f*). UMAP plotting of the 66,407 sequenced cells revealed distinct clusters (*Figure 1f–g*). To identify cell clusters, we used Louvain modulating analysis (*Joost et al., 2016*; *Wolf et al., 2018*), which resulted in 18 distinct clusters (*Figure 1g*). We assigned seven main classes of cells to the clusters. Keratinocytes represented the largest portion of cell types in the analysis (blue clusters - 0, 1, 2, 5, 8, 9, 11, 12, 17) (*Figure 1h*). We identified three distinct fibroblast clusters (green clusters - 3, 4, 16). These clusters expressed *Twist2*, *Pdgfra*, and *En1* (*Figure 1h*). We also identified other clusters based on their distinct gene expression profile (*Figure 1h*; *Supplementary file 1*), such as immune cells (cluster 6), vasculature (clusters - 7, 15), Schwann cells (cluster 10), melanocytes (cluster 13), and a pericyte population (cluster 14).

To identify a unique cell type that supports the ability to regenerate hair follicles, we quantified the number of cells for each condition within a cluster and graphed them as a percentage of cell condition within a cluster (*Figure 1i*). Unexpectedly, there was no cluster exclusively defined by over-representation of the regenerating condition. Regenerating (green) and scarring (red) conditions were found in similar proportions amongst Keratinocytes, lymphocytes/blood, melanocytes, Schwann cells, blood vessels and fibroblasts (*Figure 1i*). However, there were two clusters that were distinct based on the percentage of cell types from a specific condition. Cluster 16 consisted of 99.9% of fibroblasts from homeostatic skin, while Cluster 3 consisted of 67.8% of developing fibroblasts (*Figure 1i*). In addition, cluster 4 consisted of 25.7% of regenerating condition, 57.6% of the scarring conditions. Since fibroblasts are known to play a critical role in regulating hair follicle reformation in wounds by differentiating into de-novo DP (*Driskell et al., 2013*; *Plikus et al., 2017*; *Rognoni et al., 2016*), we chose to further analyze clusters 3, 4, and 16.

We subset the fibroblast clusters 3, 4, 16 and re-clustered them by performing an integration analysis (*Polański et al., 2020*). This allowed us to test if different fibroblast subtypes, such as the papillary and reticular/hypodermal/fascia lineages, would be represented uniquely within a condition (*Figure 1j–l*). UMAP plotting of the integrated analysis revealed 10 distinct clusters (*Figure 1k*). The largest cluster (cluster 0) expressed markers of the papillary fibroblast lineage (*Figure 1e–g*). Four clusters (clusters 1, 2, 3, 4) distinctly expressed markers of the reticular/hypodermal/fascia layers of the dermis (*Figure 1k*; *Figure 1—figure supplement 1a–b*; *Supplementary file 1*; *Correa-Gallegos et al., 2019*; *Joost et al., 2018*). Three clusters (clusters 6, 7, 8) expressed markers of the dermal papilla and dermal sheath (*Figure 1e–g*). One cluster was distinctly represented by *Acta2* indicating a myofibroblast subtype (cluster 9).

To identify conditions that are represented within specific fibroblast subtypes, we quantified the percentage of cells from each condition within the sub-clusters (*Figure 1l*). Our results revealed that four clusters were uniquely represented by four conditions. The myofibroblast sub-cluster (cluster 9) contained 69% of cells from the regenerating condition, while a hypodermal fascia cluster (cluster 1) contained 92.2% of cells from the scarring condition. Cluster 4, identified by the high expression of *Cochlin* (*Coch*), an inner ear ECM protein, contained 80.8% of cells from the homeostasis condition and a DP/DS sub-clusters (cluster 6) contained 95.0% of cells from the homeostasis condition. The

papillary fibroblast cluster (cluster 0) contained 98.1% of cells from the developing skin condition. The DP sub-cluster (cluster 7) consisted of 56.5% cells from the developing condition and 37.1% cells from the regenerating condition. These results indicate that there is a distinct separation based on condition between fibroblast sub-types neonatal papillary fibroblasts versus reticular/hypodermal/fascia. Moreover, regenerating, and scarring conditions were present in similar proportions across multiple fibroblasts subtypes, besides clusters of hypodermal and fascia cells that were mainly composed of the scarring condition. This suggested that both wounding conditions shared similar cellular and molecular signatures that may be driven by the wound healing process.

Since our goal was to identify a cell type that uniquely represented the ability to support hair follicle formation that does not include a wound induced signature, we focused on cluster 0 which are developing fibroblasts. Developing fibroblasts migrate into wounds and support the reformation of hair follicles (*Driskell et al., 2013*; *Rognoni et al., 2016*). In homeostatic skin DP and DS are critical for hair follicle growth and cycling. However, they do not migrate into wounds to support hair follicle reformation (*Johnston et al., 2013*; *Kaushal et al., 2015*; *Abbasi et al., 2020*). Myofibroblasts are a major contributor to scar formation but may not normally differentiate into dermal papilla during hair follicle regeneration (*Lim et al., 2018*; *Plikus et al., 2017*). Furthermore, reticular and fascia fibroblasts contribute to scar formation but do not differentiate into dermal papilla (*Correa-Gallegos et al., 2019*; *Driskell et al., 2013*). We conclude that the developing fibroblasts are unique and might harbor the potential to support hair follicle regeneration in wounds.

## Lef1 drives neonatal papillary lineage trajectory in developing fibroblasts

Neonatal developing skin has been extensively studied in the context of functional fibroblasts heterogeneity (*Driskell et al., 2013*). To identify fibroblasts lineages in scRNA-seq data, we isolated cell clusters from developing skin using pan fibroblast marker *Pdgfra* and *Twist2* (*Figure 2d*, *Figure 2—figure supplement 1*; *Collins et al., 2011*) and then overlaid previously identified molecular markers of different lineages including papillary (*Dpp4+/Dlk1-/Ly6a-*), reticular (*Dpp4-/Dlk1+/Ly6a+/-*) and hypodermal fibroblasts (*Dpp4-/Dlk1-/Ly6a+*) on the developing scRNA-seq UMAP (*Figure 2a–e*). We also identified small populations of fascia utilizing the new marker *Gpx3* (*Figure 2d*; *Joost et al., 2020*). Furthermore, the dermal sheath cluster was identified by expression of *Acan* and *Acta2* (*Heitman et al., 2020*; *Figure 2d*, *Figure 2—figure supplement 1*), which is plotted between the dermal papilla (DP) and papillary fibroblasts clusters (*Figure 2d–e*). The DP clusters were defined by expression of *Corin*, a well-established marker for DP (*Enshell-Seijffers et al., 2008*). Interestingly, we also found novel heterogeneity within papillary fibroblast clusters. Besides the *Dpp4+* clusters, *Lef1* and *Dkk1* expressions marked distinct papillary fibroblast clusters (*Figure 2i*, *Figure 2—figure supplement 1*). Overall, we concluded that the previously identified fibroblasts heterogeneity and the lineages could be identified in the scRNA-seq analysis of developing skin, with the additional subclusters of papillary fibroblasts (*Figure 2e*; *Driskell et al., 2013*).

To estimate and predict the future state of developing fibroblasts and to identify the driver genes for fibroblast heterogeneity, we performed RNA velocity analysis (*Figure 2f*; *Bergen et al., 2020*). RNA velocity computationally compares the ratio of spliced and un-spliced mRNA from scRNA-seq data to estimate differentiation trajectories. The overall velocity analysis of developing fibroblasts is projected as arrows on the UMAP plot and it is defined by a list of putative driver genes (*Supplementary file 2*). Velocity analysis predicts that the dermal papilla is a differentiation endpoint, while papillary fibroblasts converge into the *Dpp4* subcluster. Interestingly, the predicted differentiation trajectory of reticular fibroblasts suggests that they differentiate into arrector pili and also converge into the *Dpp4* papillary subcluster. This contradicts previous in vivo lineage-tracing studies of the Dlk1+ fibroblast populations but agrees with ex vivo studies that tested the differentiation potential of *Dpp4-/Dlk1+/Ly6a-* fibroblast populations in chamber grafting and hydrogel assays (*Driskell et al., 2013*). Hypodermal and fascia fibroblast clusters were predicted to maintain the lower lineage (*Figure 2f*).

RNA velocity analysis predicts cellular trajectories using a list of the same driver genes that are ranked differently in each cluster (*Figure 2i*, *Supplementary file 2*). These genes are ranked based on their roles in driving the velocity trajectories. We performed PANTHER pathway classification on this list genes and found that the most highly represented pathways included Integrin signaling, Wnt signaling, Cadherin signaling and TGFb signaling pathways (*Figure 2h*; *Supplementary file 5*). We

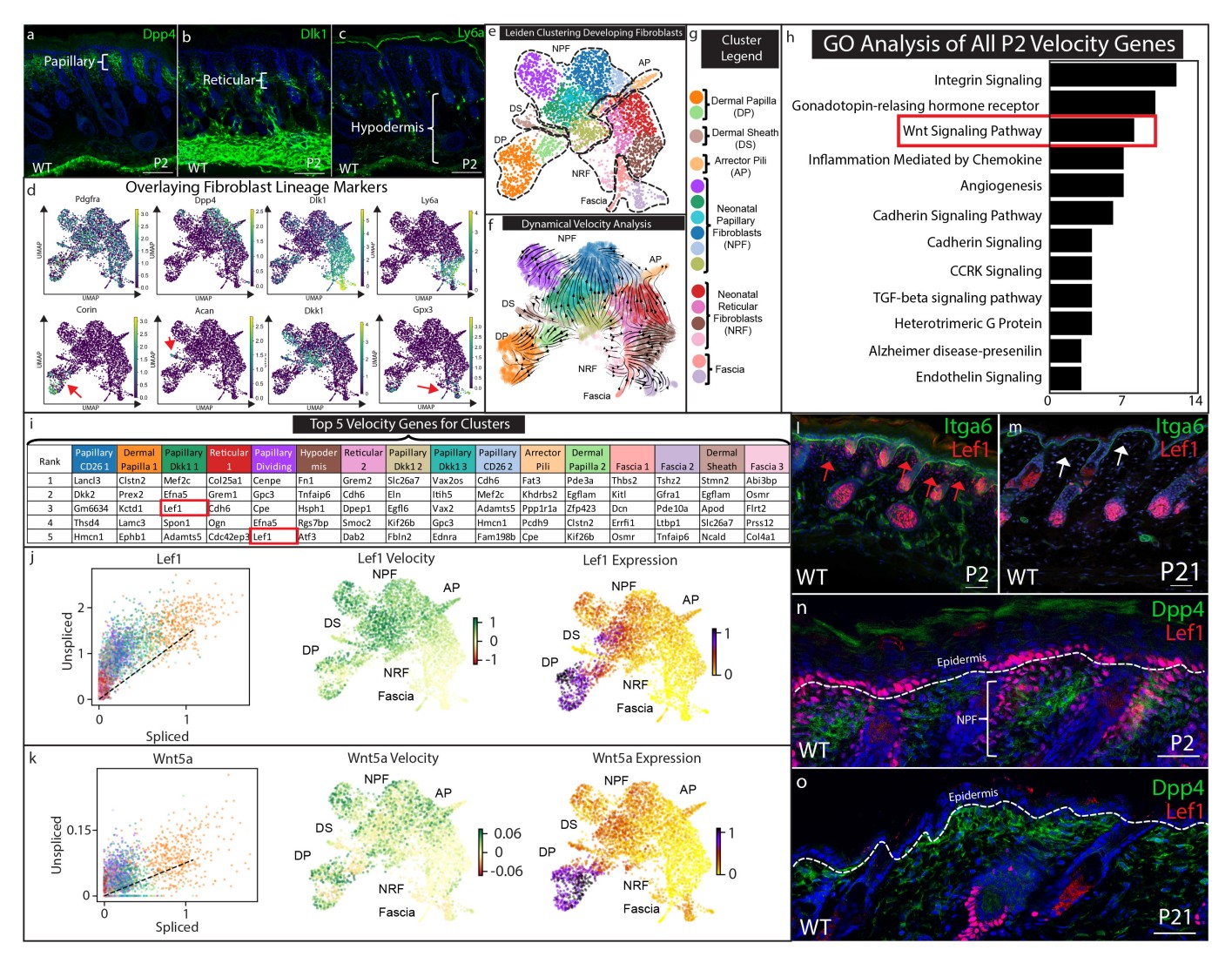

**Figure 2.** Papillary fibroblast lineage trajectory is defined by Lef1 in RNA Velocity analysis. (a–c) Immunostaining development (P2) skin for Papillary (a) (Dpp4), Reticular/Hypodermal (b) (Dlk1/Pref1), and Hypodermal makers (c) (Ly6a). (d) Overlaying fibroblast lineage markers on UMAP projections of developing fibroblasts. (e) UMAP analysis of developing fibroblasts (f–g) RNA Velocity analysis overlaid on UMAP projection of developing fibroblasts. Clusters are colored coded and labeled. (h) GO Analysis of 195 driver genes of RNA Velocity for developing fibroblasts. (i) Top 5 Velocity driver genes of developing fibroblast clusters. (j) Phase contrast analysis of *Lef1* and *Wnt5a*. (i–m) Immunostaining developing (P2) and homeostatic (P21) time points for Lef1 expression and counterstained with Itga6. Red arrows indicate areas of papillary dermis expressing Lef1. White arrows indicate papillary dermal region. (n–o) Immunostaining developing (P2) and homeostatic (P21) skin for Lef1 and Dpp4. Scale bars are 100 µm.

The online version of this article includes the following figure supplement(s) for figure 2:

**Figure supplement 1.** Classifying fibroblast clusters in developing fibroblasts.

**Figure supplement 2.** Clustering and Velocity analysis of regenerating, scarring, and developing fibroblasts.

next ranked the top five genes that drive the velocity of each cluster (*Figure 2i*). We found that *Dkk2*, *Mef2c*, *Lef1*, and *Egfl6* were identified to drive the velocity for different papillary clusters (*Figure 2i*). *Col25a1*, *Smoc2*, *Fn1*, *Dcn*, and *Col4a1* were found to drive velocity to reticular and fascia clusters. Interestingly, *Lef1* was found in the Papillary Dkk1 fibroblasts and the Dividing Papillary subclusters (*Figure 2i*). This suggests that the expression of *Lef1* may drive RNA velocity toward papillary fibroblast and dermal papilla. We further analyzed Wnt signaling genes found in the papillary and dermal papilla clusters to determine the direction of the trajectory by phase contrast

analysis (*Figure 2j*) Phase contrast analysis predicts that *Lef1* and *Wnt5a* velocity increases towards to the papillary fibroblast and dermal papilla clusters (*Figure 2j–k*).

To validate the expression and localization of *Lef1* in developing skin fibroblasts, we visualized the in vivo expression pattern via immunostaining (*Figure 2l–o*). Our results confirmed the population of neonatal papillary fibroblasts that is found only in developing skin (*Figure 2j*, red arrows) and not in juvenile homeostatic skin (*Figure 2k*, white arrows). Co-staining with *Dpp4* confirmed that *Lef1* marked a subset of neonatal papillary fibroblasts (*Figure 2l*) that was also not found in juvenile skin (*Figure 2m*).

The transient expression of *Lef1* in neonatal papillary fibroblast correlates with the ability of neonatal developing skin to regenerate hair follicles in wounds (*Rognoni et al., 2016*). In addition, velocity analysis predicts that Lef1 drives developing fibroblast velocity to papillary fibroblast and DP. Importantly, subset analysis of regenerating and scarring fibroblasts did not identify fibroblasts defined by *Lef1* expression, indicating that this transient population is unique to developing skin (*Figure 2—figure supplement 2a–f*; *Supplementary file 4*). We hypothesize that *Lef1* expression in fibroblasts marks the ability of developing skin to regenerate hair follicles during the wound healing process.

## Neonatal skin regeneration requires Lef1 expression in fibroblasts

Based on our scRNA-seq findings, we postulated that Lef1 defines a neonatal papillary fibroblast that supports skin regeneration during neonatal wound healing. To test if *Lef1* in fibroblasts is required to support neonatal regeneration, we produced a tissue-specific knockout model. We utilized the dermal specific promoter *Twist2* to drive Cre expression, bred with a mouse line with flox sites flanking Exon 7 and 8 of the *Lef1* locus (*Yu et al., 2012*). We called this mouse line Twist-Lef1cKO (*Figure 3a*). Twist-Lef1cKO mice were viable and fertile with small shifts in hair follicle phenotypes that resulted in less dense fur (*Fine et al., 2020*). We confirmed dermal *Lef1* ablation from the papillary fibroblast at P2 and from adult DP by immunostaining (*Figure 3b–e*). We also performed 2 mm wounds on P2 Twist-Lef1cKO and wild-type littermates harvesting at 7dpw (*Figure 3f*). Our analyses revealed that *Lef1* was expressed in the de-novo DP and regenerating hair follicle buds of wild-type mice (*Figure 3g,i*), but that wounded Twist-Lef1cKO mice lacked regeneration (*Figure 3h,j,k*). We conclude that *Lef1* expression in neonatal fibroblasts is required for neonatal hair follicle regeneration in wounds.

## Dermal Lef1 expression primes adult skin to enhance regeneration in large wounds

Our scRNA-seq data, together with tissue-specific knockout, implies that *Lef1* expression in the dermis could convey the regenerative ability of neonatal papillary fibroblasts in adult skin. We hypothesize that constitutively active *Lef1* expression in dermal fibroblasts will prime skin to support regeneration. To induce *Lef1* expression in the dermis of the skin, we utilized a previously published transgenic mouse model Lef1KI (*Lynch et al., 2016*) bred with Twist2-Cre mouse line called the Twist-Lef1KI line (*Figure 4a*). We verified that *Lef1* was expressed in all dermal fibroblast compartments in Twist-Lef1KI mice during development and in adult homeostatic skin conditions (*Figure 4b–e*). Importantly, the only phenotype detected was the early entry of the hair follicle cycling into anagen at P21 in Twist-Lef1KI mice (*Figure 2d–e*). In addition, Twist-Lef1KI mice have similar life spans as wild type mice without any adverse effects (*Figure 4—figure supplement 1*). Our results reveal that induced expression of *Lef1* in fibroblasts does not negatively affect the development and homeostasis of murine skin.

Adult skin normally does not regenerate but has the capacity to reform non-functional hair follicles (small and lack arrector pili) in wounds 1 cm$^2$ or larger (*Ito et al., 2007*). To test if overexpression of Lef1 in adult dermal fibroblasts enhances hair regeneration in adult skin, we wounded wild type (Lef1KI) and Twist-Lef1KI mice at P24 with 1.2 cm$^2$ wounds and harvested at 24dpw (*Figure 4f*). Our analysis comparing Lef1KI and Twist-Lef1KI wound beds revealed enhanced hair regeneration (*Figure 4f–k*). There were five times more regenerating hair follicles on average in Twist-Lef1KI wound beds compared to wild type wounds (*Figure 4j–k*). The non-functional hair follicles that typically regenerate in large adult wounds are small and lack the arrector pili, a smooth muscle that makes hair stand up when contracted. Strikingly, the hair follicles regenerating in the Twist-Lef1KI

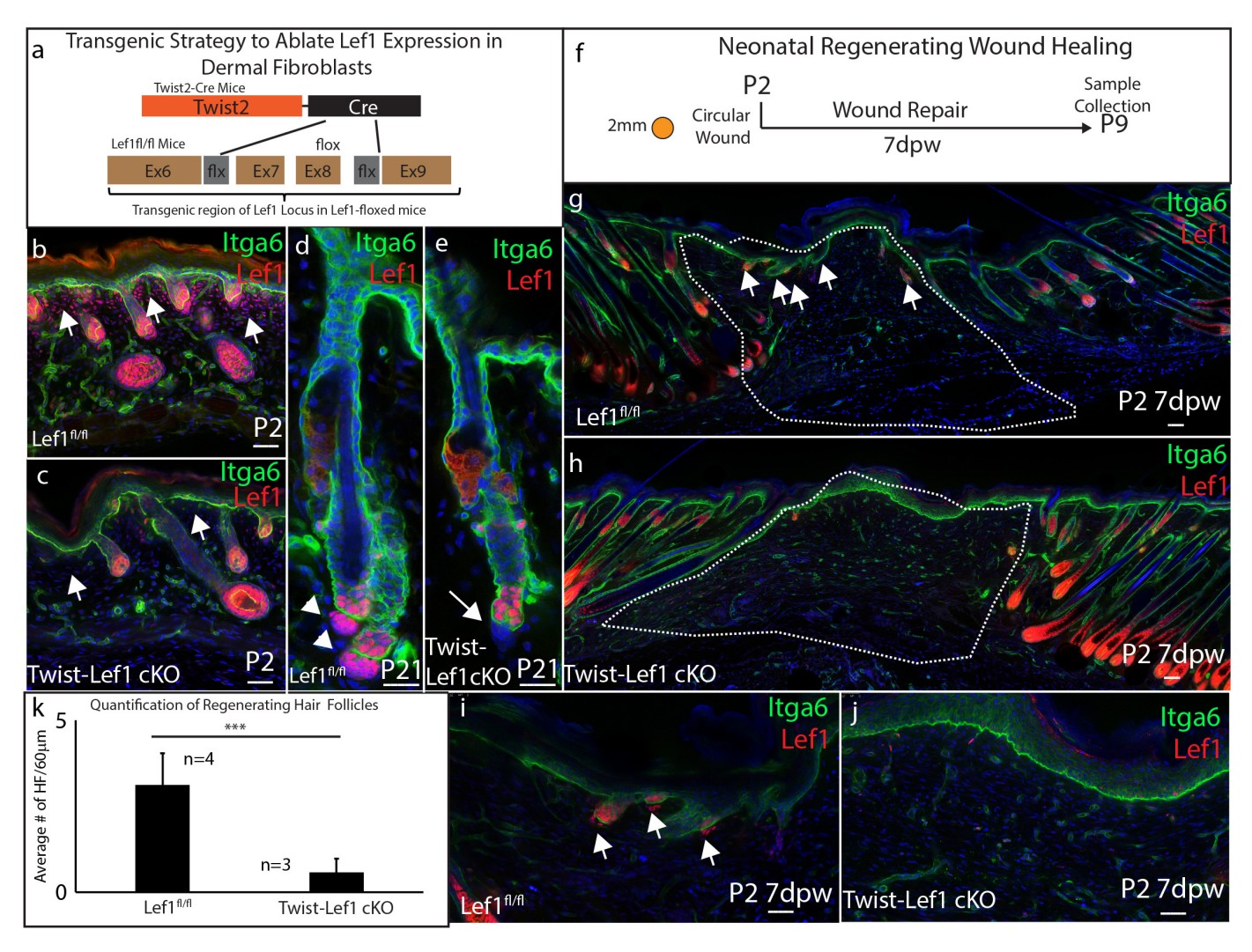

**Figure 3.** Dermal Lef1 is required to support hair follicle regeneration in neonatal wounds. (**a**) Schematic representation of transgenic strategy to ablate *Lef1* expression in dermal fibroblasts. Scale bars are 50 µm (**b–e**) Verifying dermal Lef1 ablation by immunostaining. White arrows indicate papillary regions (**b–c**) or dermal papilla (**d–e**). Scale bars are 50 µm (**f**) Schematic describing 2 mm circular wounds and harvest times to test if dermal *Lef1* is required for regeneration. (**g–i**) Immunostaining of P2 7 days post wound (7dpw) skin for Itga6 and Lef1. (g-h Scale bars are 100 µm) Wound beds are highlighted by white outline. White arrows indicate regenerating hair follicles. (**i-j** Scale bars are 50 µm) (**k**) Quantification of regenerating hair follicles in wounds beds of n = 4 Lef1fl/fl (WT) and n = 3 Twist-Lef1cKO mice. p<0.003.

wound beds contained arrector pili (*Figure 4i*) and were larger than hair follicles found in wild-type wounds. We conclude that *Lef1* expression in the dermis of the skin leads to enhances regeneration.

## Dermal Lef1 transforms scarring wounds to be regenerative

Wound healing studies utilize a standardized approach to account for the hair follicle cycle, which influences the wound healing process (*Ansell et al., 2011*). Consequently, the bulk of wound healing and regeneration studies are performed at 3 weeks of age (~P21) or at 7–8 weeks of age (~P57) (*Gay et al., 2013*; *Guerrero-Juarez et al., 2019*; *Ito et al., 2007*; *Lim et al., 2018*; *Plikus et al., 2017*; *Plikus et al., 2008*). In addition, a standard size wound of roughly 8 mm in diameter or smaller is used, which heal by scarring (*Driskell et al., 2013*; *Lim et al., 2018*; *Rognoni et al., 2016*). Since Lef1KI mice showed enhanced regenerative capacity in large wounds, we explored the regeneration

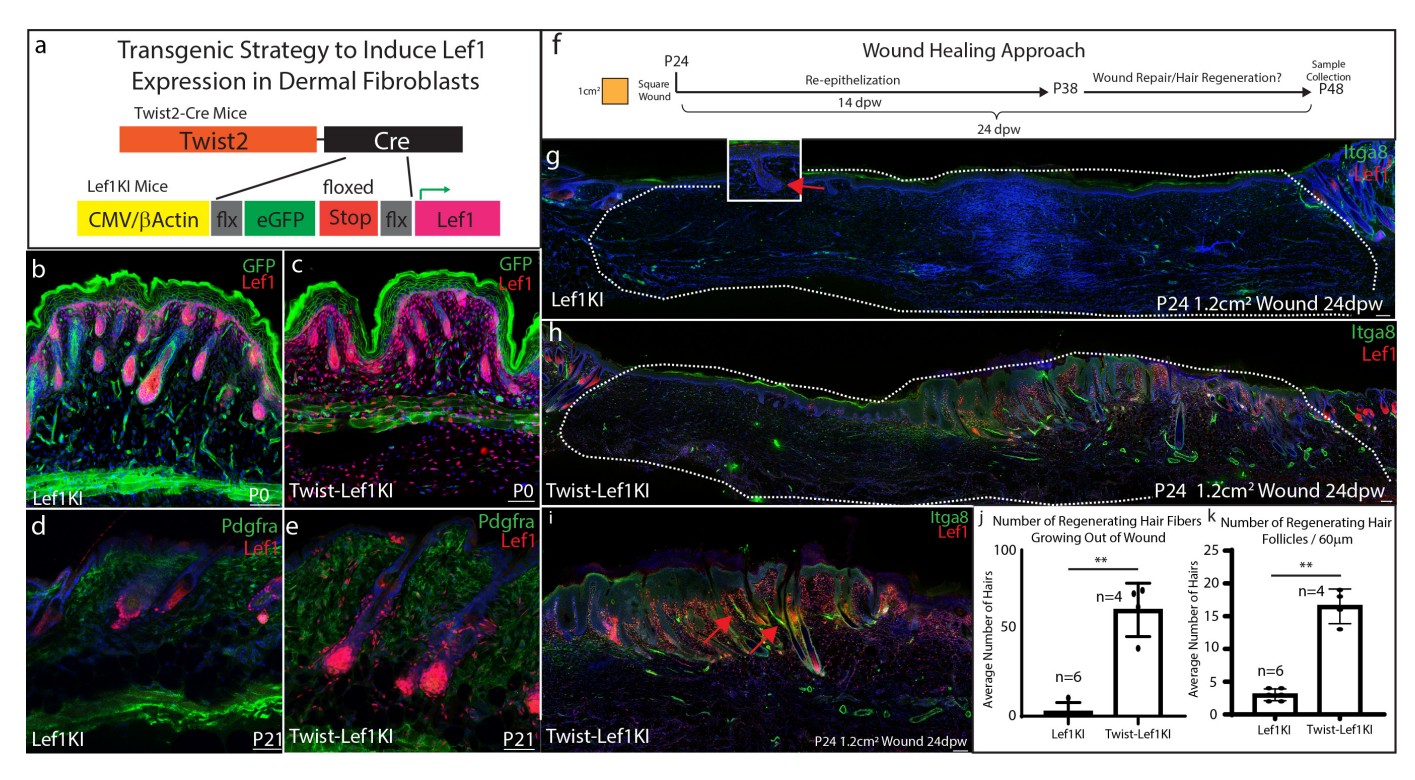

**Figure 4.** Lef1 expression in the dermis enhances skin regeneration in large wounds. (**a**) Schematic representation of transgenic strategy to ectopically express *Lef1* in fibroblasts during development, homeostasis, and wound healing. (**b–c**) Immunostaining for Lef1 and GFP in P0 Lef1KI and Twist-Lef1KI (Twist-Cre+/Lef1KI+) mice. (**d–e**) Immunostaining for Lef1 and PDGFRa in P21 Lef1KI and Twist-Lef1KI mice. (**f**) Schematic representation of wound healing approach. (**g–k**) Immunostaining tissue for Lef1 and Itga8 (arrector pili) from the wounds of Lef1KI and Twist-Lef1KI mice. Regenerating hair follicles in Lef1KI mice can be found by red arrows. Wound beds are marked by white dotted outlines. High resolution area of regenerating hair follicles in Twist-Lef1KI mice (**i**). (**j**) Quantification of regenerating hair follicles growing out of the wound beds of Lef1KI n = 6 and Twist-Lef1KI n = 4 mice p<0.004. (**k**) Quantification of regenerating hair follicles detected in 60 μm sections of wound beds. p<0.001. Scale bars at b-d are 100 μm. Scale bars at g-i are 200 μm.

The online version of this article includes the following figure supplement(s) for figure 4:

**Figure supplement 1.** Twist-Lef1KI mice does not show overt phenotypic variations.

potential of Twist-Lef1KI skin in wounds that normally scar. To determine if the hair follicle cycle influences skin regeneration we performed wounds at different time points: P24 which undergoes a hair follicle cycle during wound healing; P40 which heals during the resting phase of the hair cycle; and P90 which is the beginning of the un-synchronized hair follicle cycling that occurs in adult mice (*Figure 5a*). Scars formed in wounds performed at P24 in both WT and Twist-Lef1KI mice (*Figure 5b–c,l*). However, hair follicle regeneration with arrector pili occurred in wounds performed at P40 and P90 in Twist-Lef1KI mice (*Figure 5d–k,l*; *Figure 5—figure supplement 1*; *Figure 5—figure supplement 2*). Occasionally, we observed cyst-like structures in regenerated wounds, similar to that observed in WIHN (*Ito et al., 2007*). We conclude that *Lef1* expression in the dermis induces skin to support hair follicle regeneration in wounds with variation depends on the timepoints and hair follicle cycle.

Our wound healing studies mirror previous work suggesting that the size of a wound can dictate the potential of skin to regenerate instead of scar (*Ito et al., 2007*). To explain 'why size matters' we investigated and mapped the timing of wound closure and regeneration in the context of molecular signals within the dermal macro-environment that are dynamically changing throughout the hair follicle cycle (*Plikus et al., 2008*; *Wang et al., 2017*). Previous studies have defined the activators and

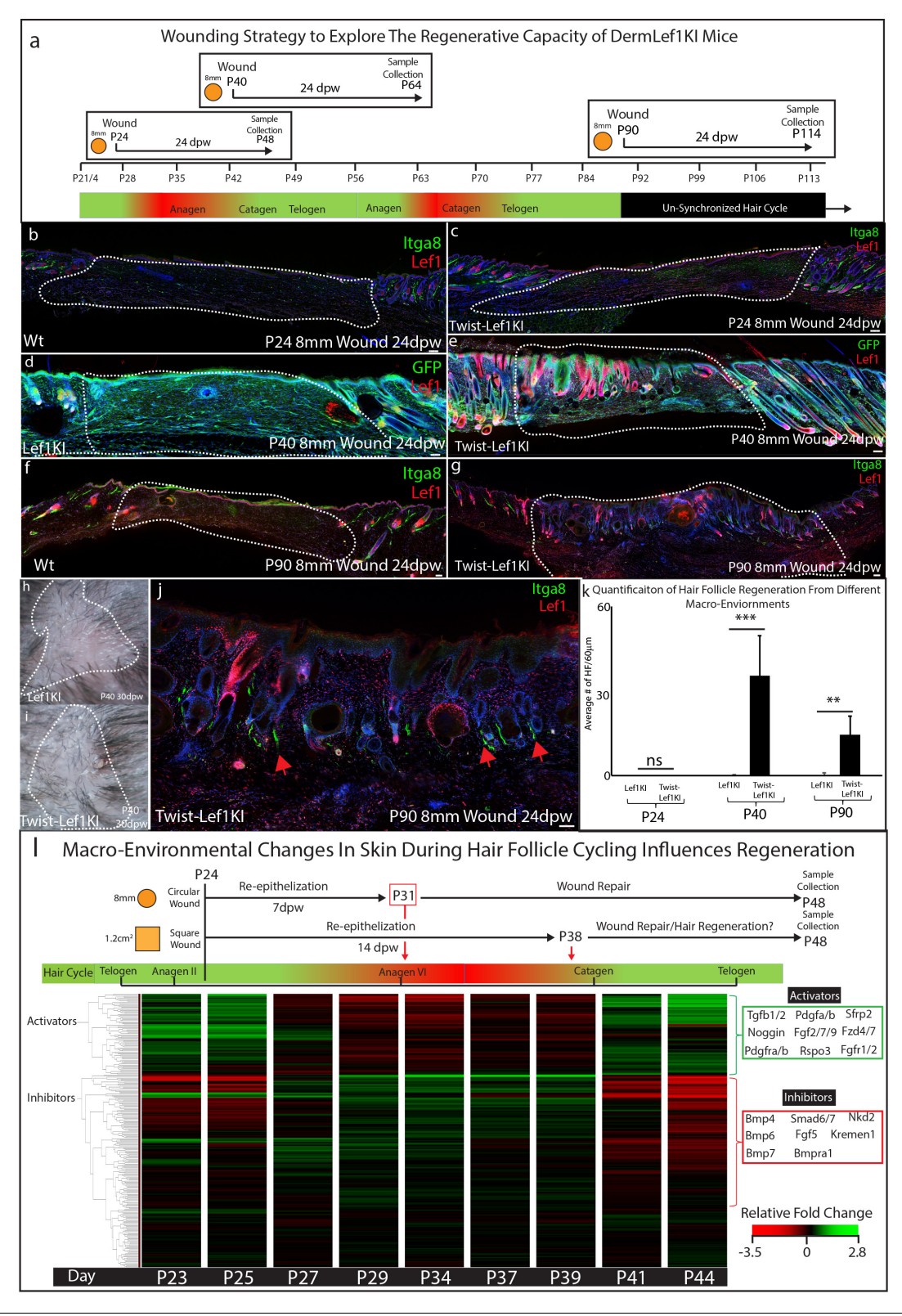

**Figure 5.** Lef1 expression in the dermis enhances skin regeneration in permissive macro environments. (**a**) Schematic of the differential wound healing analysis from different time points of the hair follicle cycle in murine skin. (**b–g,j**) Immunostaining analysis of wounds 24dpw from P24 (**b–d**), P40 (**d–e**), and P90 (**f–g,j**). Sections were stained for either Itga8 and Lef1 (**b–c,f–g,j**) or GFP and Lef1 (**d–e**). Itga8 labels arrector pili. (**h–i**) Macroscopic analysis of wound beds from WT (Lef1KI) and Twist-Lef1KI mice. Hair follicles could be seen growing out of the wound bed of Twist-Lef1KI mice. (**k**) Quantification

*Figure 5 continued*

of the average number of regenerating hair follicles detected in wounds beds of 60 µm sections from P24 24dpw (ns, WT-n = 4, Twist-Lef1KI -n = 4), P40 24dpw (p<0.0009 WT-n = 4, Twist-Lef1KI -n = 4), and P90 24dpw (p<0.007 WT-n = 3, Twist-Lef1KI -n = 4). (I) Schematic representation of different wound healing approaches above a heatmap of microarray data (*Lin et al., 2009*) from key time points during hair follicle cycling. Heatmap is generated from a list of Activator and Inhibitor genes reported to regulate the progression of the hair follicle cycle (*Wang et al., 2017*). Scale bars are 100 µm.

The online version of this article includes the following figure supplement(s) for figure 5:

**Figure supplement 1.** Analysis of the regenerative potential of Twist-Lef1KI wounds at P40.

**Figure supplement 2.** Immunostaining analysis of GFP expression in scarring and regenerating wounds of Lef1KI and Twist-Lef1KI mice.

inhibitors that regulate hair follicle cycling during this time (*Wang et al., 2017*; *Supplementary file 3*). Activators include the Wnt signaling pathway, *Noggin*, *Tgfb*, and fibroblast markers such as *Pdgfra*. Inhibitors include the BMP signaling pathway. These activators and inhibitors have been previously shown to modulate *Lef1* activity (*Jamora et al., 2003*). To explain why Twist-Lef1KI mice scar at P24, we generated a diagram of wound re-epithelization and healing time points comparing 8 mm circular wounds to 1.2 cm² wound in the context of gene expression for inhibitors and activators at all time points during the hair follicle cycle (*Figure 5I*). 8 mm circular wounds re-epithelialize 7 dpw (*Driskell et al., 2013*) in a macro-environment with high levels of Lef1 inhibitors, which are not inductive to hair follicle formation (*Figure 5I*). In contrast, large 1.2 cm² wounds re-epithelialize during the time point of the hair cycle that has high activators and low inhibitors (*Figure 5I*). We conclude that the size of a wound dictates the time point when wound closure occurs which is associated with inhibitor or inductive regenerative signals.

## Discussion

It is well established that embryonic and neonatal skin has the potential to heal in a scar-less manner (*Walmsley et al., 2015a*). However, the cell types and signals that can transfer this ability to adult skin have not been identified. In this study, we identify developing papillary fibroblasts as a transient cell population that is defined by *Lef1* expression. Inducing *Lef1* expression in the adult dermis will prime skin to support enhanced hair follicle regeneration. Interestingly, ectopic *Lef1* expression in the dermis did not result in any adverse phenotypes during development (*Supplementary file 4*).

scRNA-seq has provided an unprecedented insight in the cell types and molecular pathways that are represented within regenerating and scarring wounds (*Guerrero-Juarez et al., 2019*; *Haensel et al., 2020*; *Joost et al., 2018*; *Abbasi et al., 2020*). For example, comparisons of these different conditions have identified the activation of the Shh pathway as a key pathway to support hair follicle regeneration in wounds of different sizes (*Lim et al., 2018*). However, the activation of ectopic Shh in either the epidermis or dermis is associated with unwanted phenotypes and cancer (*Fan et al., 1997*; *Grachtchouk et al., 2000*; *Oro et al., 1997*; *Sun et al., 2020*). Unexpectedly, we discovered a neonatal papillary fibroblast population that is defined by *Lef1* expression in developing skin instead of in regenerating conditions. Importantly, our comparative analysis of regenerating and scarring wounds reveals their similarities (*Figure 1i*, *Figure 2—figure supplement 2*) more than their differences suggesting that the wound environment is a powerful driving force for gene expression even with a heterogeneous population of cells. Consequently, the basic mechanisms occurring during development and maintenance of neonatal tissue may hold the keys to transforming adult tissue to regenerate instead of scarring.

Studies involving the activation of the Wnt and beta-catenin pathways in skin have led to important discoveries in wound healing research, but have produced contrasting results in the context of fibroblast biology and hair follicle formation (*Chen et al., 2012*; *Hamburg-Shields et al., 2015*; *Ito et al., 2007*; *Lim et al., 2018*; *Mastrogiannaki et al., 2016*; *Rognoni et al., 2016*). The activation of beta-catenin in fibroblasts during wound healing has been shown to inhibit hair follicle reformation in scars (*Lim et al., 2018*; *Rognoni et al., 2016*) while deactivating Wnt signaling inhibited hair reformation (*Myung et al., 2013*). Moreover, persistent Wnt activity in the dermis has been recently shown to drive WIHN toward fibrotic wound healing (*Gay et al., 2020*). In contrast, the activation of beta-catenin during development increases hair formation and size (*Chen et al., 2012*; *Enshell-Seijffers et al., 2010*). One explanation for these conflicting results is the presence of

different fibroblast lineages present in either development or wounds, which express different Lef/Tcf factors to direct a Wnt/beta-catenin-specific response (*Driskell et al., 2013*; *Philippeos et al., 2018*; *Rognoni et al., 2016*). We and others have shown that *Lef1* is expressed in embryonic and neonatal papillary fibroblasts, which is lost in adult fibroblasts (*Driskell et al., 2013*; *Mastrogiannaki et al., 2016*; *Philippeos et al., 2018*; *Rognoni et al., 2016*). Here, we demonstrate that *Lef1* is a key transcription factor in developing fibroblasts that supports hair follicle formation in adult wounds. Our data indicates that *Lef1* expression in the dermis of the skin primes a response to support hair follicle neogenesis and arrector pili reformation in the wound. Our results align with studies that induced *Lef1* expression in cultured human dermal papilla cells to enhance hair follicle formation in vitro (*Abaci et al., 2018*). In humans, at the age of 50+, the quality of the papillary dermis gradually deteriorates, which correlates with the aging skin decreased wound healing abilities (*Haydont et al., 2019*; *Marcos-Garcés et al., 2014*). Our data suggests that activating the papillary region to retain its identity by expression of *Lef1* has the potential to enhance wound healing in humans.

# Materials and methods

**Key resources table**

| Reagent type (species) or resource | Designation | Source or reference | Identifiers | Additional information |
|---|---|---|---|---|
| Strain, strain background (*M. musculus*) | Dermo1-Cre or Twist2-Cre, C57Bl6/BalbC | Jackson Laboratory | RRID:MGI:3840442 | |
| Strain, strain background (*M. musculus*) | Lef1KI, C57Bl6/BalbC | Engelhardt lab - University of Iowa | Other | A gift from the Engelhardt lab |
| Strain, strain background (*M. musculus*) | Lef1KO, C57Bl6/BalbC | Xue lab - University of Iowa | RRID:MGI:5447957 | A gift from the Xue lab |
| Sequencing kit | Single Cell 3' Library and Gel Bead Kit v2 | 10X Genomics | Cat#: PN-120267 | |
| Antibody | Anti-Integrin alpha 8 (*Itga8*) (goat polyclonal) | R and D Systems | Cat#: AF4076 | IF(1:200) |
| Antibody | Anti-Pref1/*Dlk1* (rabbit monoclonal) | R and D Systems | Cat#: AF8277 | IF(1:200) |
| Antibody | Anti-*Dpp4*/CD26 (goat polyclonal) | R and D Systems | Cat#: AF954 | IF(1:200) |
| Antibody | Anti-*Pdgfra* (goat polyclonal) | R and D Systems | Cat#: AF1062 | IF(1:200) |
| Antibody | Anti-*Lef1* (rabbit monoclonal) | Cell Signaling | Cat#: 2230S | IF(1:200) |
| Antibody | Anti-Alpha smooth muscle actin (*Acta2*) (rabbit polyclonal) | abcam | Cat#: ab5694 | IF(1:200) |
| Antibody | Anti GFP (chicken polyclonal) | abcam | Cat#: ab13970 | IF(1:200) |
| Antibody | Anti-*Ly6a*/Sca1 (rat monoclonal) | Biolegend | Cat#: 108108 | IF(1:200) |
| Antibody | Anti-CD49f/Integrin alpha 6 (*Itga6*) (rat monoclonal) | BD Biosciences | Cat#: 555735 | IF(1:200) |

*Continued on next page*

*Continued*

| Reagent type (species) or resource | Designation | Source or reference | Identifiers | Additional information |
|---|---|---|---|---|
| Antibody | Anti-goat Alexa Fluor 488 (Donkey polyclonal) | Invitrogen | Cat#: A-11055 | IF(1:500) |
| Antibody | Anti-goat Alexa Fluor 555 (Donkey polyclonal) | Invitrogen | Cat#: A-21432 | IF(1:500) |
| Antibody | Anti-goat Alexa fluor 647 (Donkey polyclonal) | Invitrogen | Cat#: A-21447 | IF(1:500) |
| Antibody | Anti-rabbit Alexa fluor Plus 555 (Donkey polyclonal) | Invitrogen | Cat#: A32794 | IF(1:500) |
| Antibody | Anti-rabbit Alexa fluor 647 (Donkey polyclonal) | Invitrogen | Cat#: A-31573 | IF(1:500) |
| Antibody | Anti-chicken Alexa fluor 488 (Goat polyclonal) | Invitrogen | Cat#: A-11039 | IF(1:500) |
| Peptide, recombinant protein | Dispase | Corning | Cat#: 3254235 | |
| Peptide, recombinant protein | Collagenase Type I | ThermoFisher | Cat#: 17100017 | |
| Software, algorithm | Cellranger v3.0.2 | 10X Genomics | RRID:SCR_017344 | |
| Software, algorithm | scanpy | scanpy | RRID:SCR_018139 | |
| Software, algorithm | scvelo | scVelo | RRID:SCR_018168 | |

## Mouse models

All mice were outbred on a C57BL6/CBA background, with male and female mice used in all experiments. The following transgenic mouse lines were used in this study, Twist2-Cre (RRID:MGI: 3840442) (Cat# 008712). The ROSA26-CAG-flox-GFP-STOP-flox-Lef1 and Lef1-flox-Exon7,8-flox mice (RRID:MGI:5447957) were a kind gift and have been previously described (*Lynch et al., 2016*; *Yu et al., 2012*). Wild-type mice utilized in scRNA-seq experiments were an outbred background of C57BL6/CBA mice.

## Wounding experiments

All wounding experiments were done in accordance with the guidelines from approved protocols Washington State University IACUC. Mice were wounded at post-natal day (P) 2, P24, P40 and P90 as stated in the results section. All wounding experiments were carried out with mice under anesthetization. The dorsal hair was shaved, and wounding areas were disinfected. For full-thickness circular wound, 2 mm and 8 mm punch biopsy and surgical scissors were used to create circular wounds on P2 and P24/P40 mice, respectively. For large wounds, 1.44 cm$^2$ (1.2 × 1.2 cm) squares were excised using surgical scissors. Wounds were harvested at 7 dpw, 14 dpw, 24 dpw, and 30 dpw.

## Horizontal whole mount

This procedure was described in *Salz and Driskell, 2017*. Briefly, full thickness skin samples were collected and fixed in 4% Paraformaldehyde (PFA), before being frozen in OCT compound in cryo-block. Samples were sectioned using cryostat at 60 micro m thickness. Skin sections were then stained with primary antibodies at 4°C overnight in PB buffer and with secondary antibodies in PB buffer for 1 hr. Samples were mounted on glass cover slip with glycerol as mounting medium.

## Generation of single-cell RNA-sequencing data

Tissues were collected from dorsal regions of unwounded P2 and P21 mice, and right on the edge of 2 mm wounds 7 days post wound from P2 wound and P21 wound. Cells isolation procedure was previously described in Jensen and Driskell 2009 Nature Protocols (*Jensen et al., 2010*). Single-cell cDNA libraries were prepared using the 10X Genomics Chromium Single Cell 3' kit. For each condition, three mice were used to generate three separate libraries, making it a total of 12 libraries across all four conditions. All samples were processed individually. The prepared libraries were sequenced on Illumina HiSeq 4000 (100 bp Paired-End). Demultiplexed Paired-End fastq files were aligned to mm10 reference genome using 10X Genomics CellRanger function (RRID:SCR_017344) (Cellranger version 3.0.2). FASTQ files for this manuscript can be access at NCBI Geo-Datasets: GSE153596. The outputs of CellRanger alignment were used to create loom files by Velocyto 'run10x' function (*La Manno et al., 2018*). Loom files are the preferred input data for scVelo package (RRID:SCR_018168) to perform RNA Velocity analysis (*Bergen et al., 2020*).

## Single-cell RNA-seq data preprocessing

For basic filtering of our data, we filtered out cells have expressed less than 200 genes and genes that are expressed in less than three cells. To ensure that the data is comparable among cells, we normalized the number of counts per cell to 10,000 reads per cell as suggested by the Scanpy pipeline (RRID:SCR_018139) (*Wolf et al., 2018*). Data were then log-transformed for downstream analysis and visualization. We also regressed out effects of total counts per cell and the percentage of mitochondrial genes expressed, then scaled the data to unit variance with the maximum standard deviation 10.

## Single-cell RNA-seq data analysis

Neighborhood graph of cells was computed using PCA presentation (n PCs = 40, n neighbors = 10). The graph was embedded in two dimensions using Uniform Manifold Approximation and Projection (UMAP) as suggested by Scanpy developers. Clusters of cell types were defined by Louvain and Leiden method for community detection on the generated UMAP graph at resolution of 0.2. The analysis pipeline is published on the Driskell lab Github page as Jupyter Notebook (https://github.com/DriskellLab/Priming-Skin-to-Regenerate-by-Inducing-Lef1-Expression-in-Fibroblasts-; copy archived at https://github.com/elifesciences-publications/Priming-Skin-to-Regenerate-by-Inducing-Lef1-Expression-in-Fibroblasts-; *DriskellLab, 2020*).

## Microarray analysis

The microarray data were previously published (GEO GSE11186) (*Lin et al., 2009*). RMA normalization was used, and the bottom twentieth percentile of genes were excluded from the analysis.

## Acknowledgements

This work was supported by NIAMS R56 AR073778-01A1 and a WSU New Faculty Seed Grant 2428–9926. The authors acknowledge Sean Thompson, Jonathan Jones, Kai Kretzschmar, and Klaas Mulder for their discussion and help.The authors acknowledge the Washington State University Franceschi Microscopy Imaging Center for the continued support in microscopic imaging, and Jared Brannan for generation of our web resource.

## Additional information

### Funding

| Funder | Grant reference number | Author |
|---|---|---|
| National Institute of Arthritis and Musculoskeletal and Skin Diseases | 1 R56 AR073778-01A1 | Ryan R Driskell |
| WSU New Faculty Seed | 133408 | Ryan R Driskell |

The funders had no role in study design, data collection and interpretation, or the decision to submit the work for publication.

### Author contributions

Quan M Phan, Conceptualization, Data curation, Formal analysis, Validation, Investigation, Visualization, Writing - original draft, Writing - review and editing; Gracelyn M Fine, Lucia Salz, Conceptualization, Resources; Gerardo G Herrera, Software, Methodology; Ben Wildman, Resources; Iwona M Driskell, Resources, Project administration, Writing - review and editing; Ryan R Driskell, Conceptualization, Formal analysis, Supervision, Funding acquisition, Methodology, Writing - original draft, Writing - review and editing

### Author ORCIDs

Ryan R Driskell (iD) https://orcid.org/0000-0001-7673-2564

### Ethics

Animal experimentation: This study was performed in strict accordance with recommendations in the Guide for the Care and Use of Laboratory Animals of the National Institutes of Health. All of the animals were handled according to approved institutional animal care and use committee (IACUC) protocols (#5033/6724 and #5002/6723) of Washington State University.

### Decision letter and Author response

Decision letter https://doi.org/10.7554/eLife.60066.sa1
Author response https://doi.org/10.7554/eLife.60066.sa2

## Additional files

### Supplementary files

• Supplementary file 1. Markers used to identify cell populations in scRNA-seq analysis. Genes utilized to identify clusters in scRNA analysis.

• Supplementary file 2. Velocity genes for all developing fibroblast populations. The genes driving Velocity Trajectory in developing fibroblasts for all clusters.

• Supplementary file 3. Inhibitory genes during hair follicle cycling. List of previously identified genes hypothesized to be inhibitory during hair follicle cycle (*Wang et al., 2017*). The list also includes the relative expression values of microarray analysis (*Lin et al., 2009*).

• Supplementary file 4. Differential expression of all regenerating fibroblasts compared to scarring fibroblasts. This is a list of genes upregulated in regenerating or scarring fibroblasts when compared to each other in our SCANPY analysis.

• Supplementary file 5. Panther pathway analysis of velocity genes from developing fibroblasts. List of pathways generated from velocity genes of developing fibroblasts.

• Transparent reporting form

## Data availability

We have generated a web-resource to easily query our large datasets: https://skinregeneration.org/ We have uploaded our data to GEO-Datasets: GSE153596 Our source code can be found on our Github webpage: https://github.com/DriskellLab/Priming-Skin-to-Regenerate-by-Inducing-Lef1-Expression-in-Fibroblasts- (copy archived at https://github.com/elifesciences-publications/Priming-Skin-to-Regenerate-by-Inducing-Lef1-Expression-in-Fibroblasts-).

The following dataset was generated:

| Author(s) | Year | Dataset title | Dataset URL | Database and Identifier |
|---|---|---|---|---|
| Phan QM, Fine G, Salz L, Herrera GG, Wildman B, Driskell IM, Driskell RR | 2020 | Lef1 expression in fibroblasts maintains developmental potential in adult skin to regenerate wounds | http://www.ncbi.nlm.nih.gov/geo/query/acc.cgi?acc=GSE153596 | NCBI Gene Expression Omnibus, GSE153596 |

The following previously published dataset was used:

| Author(s) | Year | Dataset title | Dataset URL | Database and Identifier |
|---|---|---|---|---|
| Lin KK, Kumar V, Geyfman M, Chudova D, Ihler AT, Smyth P, Paus R, Takahashi JS, Andersen B | 2009 | Expression profiling of mouse dorsal skin during hair follicle cycling | https://www.ncbi.nlm.nih.gov/geo/query/acc.cgi?acc=GSE11186 | NCBI Gene Expression Omnibus, GSE11186 |

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
