## [Decision Letter]

**Acceptance summary:**

Overall this is an interesting paper whose ssRNA seq dataset and experimental analysis of phenotypes provides a valuable resource for investigating gene expression differences associated with key phases of skin development and repair. The enhancement of HF regeneration upon Lef1 overexpression is a striking result and will be of general interest to many fields including developmental, stem cell, and epithelial biologists. The work is well conducted, the results are new, and significant for skin wound healing and HF regeneration, and in sum a good fit for *eLife*.

**Decision letter after peer review:**

Thank you for submitting your article "Lef1 expression in fibroblasts maintains developmental potential in adult skin to regenerate wounds" for consideration by *eLife*. Your article has been reviewed by three peer reviewers, and the evaluation has been overseen by a Reviewing Editor and Kathryn Cheah as the Senior Editor. The reviewers have opted to remain anonymous.

The reviewers have discussed the reviews with one another and the Reviewing Editor has drafted this decision to help you prepare a revised submission.

Summary:

Overall this is an interesting paper whose ssRNA seq dataset and experimental analysis of phenotypes provides a valuable resource for investigating gene expression differences associated with key phases of skin development and repair. The enhancement of HF regeneration upon Lef1 overexpression is a striking result and will be of general interest to many fields including developmental, stem cell, and epithelial biologists. The work is well conducted, the results are new, and significant for skin wound healing and HF regeneration, and in sum a good fit for *eLife*.

Essential revisions:

The overall tone of all reviewers is enthusiastic and favorable, however with very important points raised:

1) Dermo1-Cre seems not specific to fibroblasts (and it is non-inducible). Ideally this should be addressed by using an inducible/more specific Cre mouse line. However, as the enhancement of HF regeneration is an exciting finding by itself and a new mouse model is likely out of scope of a revision, this point could be addressed textually by changing the conclusions to reference stromal cells instead of fibroblasts specifically.

2) The interpretation of the scRNA data should be bolstered with additional analyses. It is important for the authors to revisit the data and figures (including making some improved analysis), and carefully state the actual results and conclusions supporting their claims and following next steps in the manuscript.

a) ScRNA-seq analysis was superficial in relation to regeneration versus repair, especially comparison of the time points that model regeneration and scarring. Does velocity analysis predicting Lef1, or other genes, driving differentiation of one population of fibroblasts into a papillary fibroblast or DP-like state? Do multiple fibroblast subsets follow this trajectory? How do these finding compare between the two wounding time points? Does gene ontology suggest differences within one subcluster of fibroblasts between two conditions or are the major differences in the gene expression profile/function associated with each subcluster? A more complete analysis of this could shed more light on the involvement of fibroblast lineages in regenerative versus reparative healing.

b) From the ssRNA seq analysis the authors state "we identify Developing papillary fibroblasts as a transient cell population that is defined by Lef1 expression.", but this is not clear from the ssRNA seq analysis. In Figure 2—figure supplement 1, Lef1 expression seems to be largely excluded from cells within the Dpp4 expression cluster (cluster 2), and Dkk1 (Cluster 0), which define the major papillary FB clusters. Can the authors expand upon how the Velocity Analysis identifies different genes than overlaying relative expression levels on the UMAPS?

3) Surrounding the claim of a transient papillary fibroblast population (which is an important part in their paper), several parts are unclear;

I.e. they could/should explore the fibroblast populations of all conditions to compare regeneration vs. scaring and regeneration vs. development (e.g. R2Q2, R3Q5).

which of the two papillary fibroblast population(s) is/are transient? How to explain the rather minor overlap of Lef1 expression with these two papillary fib populations? Where are the two populations in situ in developing and in regenerating skin?

4) Given that the WIHN generates a significant amount of cysts, the authors have to down-tone their statement of "without adverse phenotype". As the authors also refer to Hedgehog-pathway induced de novo HF formation (a model giving rise to tumors and new HFs), they likely meant that their model does not induce apparent tumors (the cysts look different compared to the obvious BCC-like lesions trough Hh-pathway activation) – however the authors totally neglect the fact (don't mention) that the mice apparently develop cysts in addition to HFs in wounds.

Figure 5E, G, J. The regenerated HFs appear very abnormal and cyst-like. The authors state several times in the paper that Lef1 overexpression enhances regeneration without other adverse phenotypes, but these regenerated structure are very abnormal. Are they cancerous? P90 wounds appear to generate a significant amount of cysts; is this representative for all conditions or something more specific for the P90 timepoint?

---

## [Author Response]

Essential revisions:The overall tone of all reviewers is enthusiastic and favorable, however with very important points raised:1) Dermo1-Cre seems not specific to fibroblasts (and it is non-inducible). Ideally this should be addressed by using an inducible/more specific Cre mouse line. However, as the enhancement of HF regeneration is an exciting finding by itself and a new mouse model is likely out of scope of a revision, this point could be addressed textually by changing the conclusions to reference stromal cells instead of fibroblasts specifically.

We thank the reviewers for this comment and have addressed the concerns by changing the text as suggested.

2) The interpretation of the scRNA data should be bolstered with additional analyses. It is important for the authors to revisit the data and figures (including making some improved analysis), and carefully state the actual results and conclusions supporting their claims and following next steps in the manuscript.

The below points by the reviewers have been helpful in revising our manuscript and we have addressed the points as stated below. In short, we have rewritten portions of the text in the manuscript that refers to Figure 1 and Figure 2.

a) ScRNA-seq analysis was superficial in relation to regeneration versus repair, especially comparison of the time points that model regeneration and scarring. Does velocity analysis predicting Lef1, or other genes, driving differentiation of one population of fibroblasts into a papillary fibroblast or DP-like state? Do multiple fibroblast subsets follow this trajectory? How do these finding compare between the two wounding time points? Does gene ontology suggest differences within one subcluster of fibroblasts between two conditions or are the major differences in the gene expression profile/function associated with each subcluster? A more complete analysis of this could shed more light on the involvement of fibroblast lineages in regenerative versus reparative healing.

The reviewers have suggested an analysis that we have now performed and present as new Figure 2—figure supplement 2. This comparison is represented by an integrated clustering analysis mapped by marker genes and a velocity analysis. We utilized a combination of markers to create a cell atlas of each UMAP. This included Corin, Acan, Acta2, Crabp1, Mest, and Plac8.

Importantly, our velocity analysis does not predict Lef1 to drive differentiation of fibroblasts to papillary or DP like state, shown in our new Figure 2—figure supplement 2A-C. Overall dermal papilla from regenerating wounds appeared to be separated from the main population with a trajectory that might be connected to the main population. The upper fibroblast population from regenerating wounds clusters closely with the upper fibroblast population of scarring wounds, but does not show an overall integrating velocity (Figure 2—figure supplement 2B).

In this manuscript we have focused on exploring the Developing skin condition for regenerative molecular signals. However, we performed a differential expression on all cells within regenerating and scarring conditions, which revealed differences gene ontology and key genes between points. Particularly the expression of Gpx3, a fascia marker being upregulated in scarring wounds. This is now a new Figure 2—figure supplement 2.

b) From the ssRNA seq analysis the authors state "we identify Developing papillary fibroblasts as a transient cell population that is defined by Lef1 expression.", but this is not clear from the ssRNA seq analysis. In Figure 2—figure supplement 1, Lef1 expression seems to be largely excluded from cells within the Dpp4 expression cluster (cluster 2), and Dkk1 (Cluster 0), which define the major papillary FB clusters. Can the authors expand upon how the Velocity Analysis identifies different genes than overlaying relative expression levels on the UMAPS?

We thank the reviewer for allowing us to address this point. In the manuscript, we have discovered heterogeneity within neonatal papillary fibroblasts. We identified a population that express Lef1 and Dkk1 (Figure 2D) different from the CD26+ fibroblasts.

RNA velocity computes the velocity trajectories of cells in scRNA-seq experiment using the unspliced/spliced ratio of mRNA of different genes across multiple cells. The ranked list of driver genes defined by RNA velocity represented the top genes that drive the calculated trajectories for each cluster. These genes are found based on the transcriptional splicing kinetic and are different from the relative expression level.

3) Surrounding the claim of a transient papillary fibroblast population (which is an important part in their paper), several parts are unclear;

We thank the reviewer for this comment and try to address this as follows. We want to make clear that ‘transient papillary fibroblast populations’ refers to the population of papillary fibroblasts found only in Developing skin between P0 and P7 (Rognoni et al., 2016). These fibroblasts express high levels of Lef1 and Dkk1 and low levels of Dpp4/CD26 (cluster Papillary Dkk1 1 in Figure 2D, E, G and I).

I.e. they could/should explore the fibroblast populations of all conditions to compare regeneration vs. scaring and regeneration vs. development (e.g. R2Q2, R3Q5).

We have now done this. This part of the new Figure 2—figure supplement 2. We did not detect Dkk1 and Lef1 positive papillary fibroblast populations in any condition besides Developing skin.

Which of the two papillary fibroblast population(s) is/are transient? How to explain the rather minor overlap of Lef1 expression with these two papillary fib populations? Where are the two populations in situ in developing and in regenerating skin?

The transient population of papillary fibroblasts express Lef1 with a low level of Dkk1 with low levels of CD26. We have confirmed the location and the timepoint of this population by Immunofluorescent staining in Figure 2I-O. These fibroblasts located in the upper most region of the dermis and can only be found in Developing skin.

4) Given that the WIHN generates a significant amount of cysts, the authors have to down-tone their statement of "without adverse phenotype". As the authors also refer to Hedgehog-pathway induced de novo HF formation (a model giving rise to tumors and new HFs), they likely meant that their model does not induce apparent tumors (the cysts look different compared to the obvious BCC-like lesions trough Hh-pathway activation) – however the authors totally neglect the fact (don't mention) that the mice apparently develop cysts in addition to HFs in wounds.Figure 5E, G, J. The regenerated HFs appear very abnormal and cyst-like. The authors state several times in the paper that Lef1 overexpression enhances regeneration without other adverse phenotypes, but these regenerated structure are very abnormal. Are they cancerous? P90 wounds appear to generate a significant amount of cysts; is this representative for all conditions or something more specific for the P90 timepoint?

We thank the reviewer for this suggestion and have edited the manuscript as suggested. Regarding the cancerous potential of the cysts, we did not fully study the phenotypes of the cysts and hope that our toned down comments will leave this area of research for future studies.